# Can Thyroid Screening in the First Trimester Improve the Prediction of Gestational Diabetes Mellitus?

**DOI:** 10.3390/jcm11133916

**Published:** 2022-07-05

**Authors:** Zagorka Milovanović, Dejan Filimonović, Ivan Soldatović, Nataša Karadžov Orlić

**Affiliations:** 1Clinic for Gynaecology and Obstetrics Narodni Front, Kraljice Natalije 62, 11000 Belgrade, Serbia; dejan.filimonovic@gmail.com (D.F.); orlicmail@gmail.com (N.K.O.); 2Faculty of Medicine, University of Belgrade, Dr Subotica Starijeg 8, 11000 Belgrade, Serbia; soldatovic.ivan@gmail.com; 3Institute of Medical Statistics and Informatics, Dr Subotica Starijeg 15, 11000 Belgrade, Serbia

**Keywords:** gestational diabetes mellitus, first-trimester pregnancy screening, TSH, thyroid antibodies, anti TPO Ab, anti Tg Ab

## Abstract

This study aimed to evaluate the clinical utility of the subclinical hypothyroidism (SCH) marker, elevated thyroid-stimulating hormone (TSH) and thyroid antibodies in their ability to predict subsequent gestational diabetes mellitus (GDM). In a prospective clinical trial, 230 pregnant women were screened for thyroid function during the first trimester of pregnancy. Increased TSH levels with normal free thyroxine (fT4) were considered SCH. The titers of thyroid peroxidase antibody (anti TPO Ab) at >35 IU/mL and thyroglobulin antibody (anti Tg Ab) at >115 IU/mL were considered as antibodies present. According to the OGTT results, the number of pregnant women with GDM showed the expected growth trend, which was 19%. Two groups of pregnant women were compared, one with GDM and the other without. Increased TSH levels and the presence of thyroid antibodies showed a positive correlation with the risk of GDM. TSH levels were significantly higher in pregnant women with GDM, *p* = 0.027. In this study, 25.6% of pregnant women met the diagnostic criteria for autoimmune thyroiditis. Hashimoto’s thyroiditis was significantly more common in GDM patients, *p* < 0.001. Through multivariate logistic regression, it was demonstrated that patient age, TSH 4 IU/mL, and anti TPO Ab > 35 IU/mL are significant predictors of gestational diabetes mellitus that may improve first-trimester pregnancy screening performance, AUC: 0.711; 95% CI: 0.629–0.793.

## 1. Introduction

First-trimester screening for chromosomal anomalies is a milestone in pregnancy monitoring. The fact that screening with a combination of fetal ultrasound, maternal serum free-beta-human chorionic gonadotropin (β-hCG) and pregnancy-associated plasma protein-A (PAPP-A) can identify 90% of fetuses with trisomy 21 and other major chromosomal abnormalities with a false-positive rate of 5% [1] has drawn doctors’ and researchers’ attention to the possibility of screening for other pregnancy-related conditions. Scientists worldwide have concentrated on finding the first-trimester serum markers that predict other pregnancy complications. Recent studies have shown that certain maternal conditions may be predicted in the first trimester. Screening a combination of maternal factors and measuring mean arterial pressure, uterine pulsatility index, and serum placental growth factor may predict 90% of early preeclampsia cases, before 32 weeks, and 75% of preterm preeclampsia cases, before 37 weeks, with a 10% false-positive rate [1].

Thyroid dysfunction and diabetes mellitus are the two most common endocrine disorders that occur during pregnancy [2]. Gestational diabetes mellitus (GDM) is defined as carbohydrate intolerance resulting in hyperglycemia that occurs for the first time during pregnancy [3]. International Diabetes Federation (IDF) data from 2021 indicated that the prevalence of hyperglycemia in pregnancy is 16.7%, 80.3% of which is gestational diabetes mellitus, 14% in total [4]. According to the latest recommended diagnostic criteria of the World Health Organization (WHO) and the International Association of Diabetes and Pregnancy Study Groups (IADPSG) from 2013 for hyperglycemia in pregnancy, it was expected that the GDM prevalence would rise significantly [5,6]. Pregnancy hyperglycemia is associated with numerous short- and long-term complications in the mother, fetus, and neonate [7,8]. Early diagnosis and treatment of GDM, may significantly reduce the frequency and severity of perinatal complications.

Thyroid dysfunction is the second most common endocrine disorder in women, with an incidence of about 4% in the pregnant population [9], with hypofunction being significantly more prevalent. Subclinical hypothyroidism (SCH) is defined as elevated thyroid-stimulating hormone (TSH) with normal thyroxine (T4) and triiodothyronine (T3) concentrations. According to available data from current literature, hypothyroidism prevalence in pregnant women is 2.2–5% [10,11,12], with subclinical hypothyroidism occurring in 3–13% [13,14,15]. depending on the area and population studied. In patients with sufficient iodine intake, the most common cause of hypothyroidism is autoimmune thyroiditis (Hashimoto thyroiditis) [16]. The frequency of antithyroid antibodies, thyroid peroxidase antibody-anti TPO Ab or thyroglobulin antibody-anti Tg Ab in euthyroid pregnant women, according to different authors, is about 10% (2–18%) [17,18,19], while the antibodies can be found in 30–60% of hypothyroid pregnant women [16]. Thyroid disorder in pregnancy is associated with an increased risk of adverse pregnancy outcomes and pregnancy complications [11,20,21]. Subclinical hypothyroidism, if not treated, may increase the risk of miscarriage, early intrauterine fetal death [10,22], pregnancy anemia, hypertension, preeclampsia [23], postpartum hemorrhage, placental abruption [11,24] and gestational diabetes mellitus [25].

The thyroid hormones directly regulate insulin secretion and glucose metabolism. Hyperthyroidism is associated with decreased insulin sensitivity and insulin resistance, which impairs glucose homeostasis and leads to hyperglycemia. There is a link between thyroid dysfunction and diabetes mellitus. Several studies have shown an increased prevalence of thyroid disorders in diabetes mellitus patients and vice versa [26]. Both disorders are part of polyglandular autoimmune syndrome (PAS). The autoimmune process causes an irreversible loss of function, while chronic autoimmune aggressions can simultaneously modify physiological processes in the affected tissue and lead to altered organ function. Early detection of specific autoantibodies and latent organ-specific dysfunction is advocated to alert physicians to take appropriate action to prevent PAS [27]. According to the available literature, studies on thyroid dysfunction and GDM incidence in different populations have shown a connection between the two entities, although the results are not consistent [26,28,29,30,31,32]. As thyroid and pancreatic dysfunction may be linked by the same autoimmune process, the presence of autoimmune antithyroid antibodies in the first trimester may predict impaired glucose tolerance later in pregnancy.

This study aimed to examine the association of subclinical hypothyroidism and the presence of thyroid autoantibodies in the first-trimester pregnancy with impaired glucose metabolism and to determine the potential predictive value of the elevated thyroid-stimulating hormone and the presence of antithyroid antibodies in gestational diabetes mellitus screening. Existing screening tests for gestational diabetes mellitus use maternal characteristics, family, and personal history to assess the risk of one specific condition [1]. The possibility of using screening markers of thyroid function in the early first trimester of pregnancy to assess the risk of gestational diabetes mellitus might allow one test to identify the risk for two clinically important conditions, enabling appropriate and specific treatment for pregnant women at risk.

## 2. Materials and Methods

### 2.1. Study Setting and Patients

A prospective clinical study was conducted at the University Clinic for Gynaecology and Obstetrics “Narodni Front” in Belgrade from November 2019 to November 2021, among patients on antenatal control. This research was approved by the Ethical Board of the Faculty of Medicine, Belgrade University, No. 1550/XI-39.

Included in the study were pregnant women over 18 years of age with a singleton pregnancy. Multiparity, thyroid dysfunction diagnosed before pregnancy, overt diabetes mellitus, congenital malformations of the fetus, fetal death before the planned oral glucose tolerance test, and incomplete documentation were the exclusion criteria.

The minimum number of respondents for estimating the prevalence of gestational diabetes in pregnant women, calculated based on literature data indicating a 1–14% prevalence of gestational diabetes (an assumption that in our population (in Serbia) is about 10%), with an alpha error of 0.05 and an accuracy of 5%, was 138. Considering the possibility of 10% missing data, the required minimum number of respondents for this research was 152. The study enrolled 260 patients. Twenty-five patients were excluded because of incomplete documentation, one withdrew owing to early fetal demise (miscarriage), and four patients left the study. The trial covered 230 patients.

### 2.2. Procedures

All the pregnant women were followed-up every 4 to 6 weeks, had at least five to six visits and underwent thyroid function screening in the first trimester and an oral glucose tolerance test (oGTT) in the late second or early third trimester of pregnancy. The first visit occurred between 5 and 11 weeks of pregnancy to determine the exact date of the pregnancy, perform the viability ultrasound scan, obtain data on the mother and her family history, perform weight and height measurements, and administer a thyroid function screening test. At the second visit, between 11 and 14 gestational weeks, first-trimester screening for chromosomal abnormality was carried out to assess the risk of chromosomal abnormality, according to the standards of the Fetal Medicine Foundation as part of the international and national guidelines, and to detect major congenital anomalies as early as possible. A detailed anatomy ultrasound scan between 20 and 24 gestational weeks was carried out on every pregnant woman as the standard of pregnancy monitoring, to check growth and exclude structural anomalies. The oGTT with 75 g, was performed between 24 and 30 gestational weeks. Growth and well-being ultrasound scans were performed between 28 and 32 gestational weeks and were repeated between 36 and 38 gestational weeks if the baby was not delivered earlier.

On her first visit to the high-risk pregnancy department, the pregnant woman’s medical history was taken and stored in the hospital’s electronic database ZIS (Zdravstveni Informacioni Sistem–Health information system). This information included date of birth, last menstrual period, history of chronic diseases, medication used, surgeries, previous pregnancy outcomes, births and miscarriages, and a family history of thyroid disease and diabetes mellitus (first and second-line relatives). The body height and weight of pregnant women were measured and used to calculate the body mass index (BMI) at the pregnancy’s start, with the standard formula weight (kg)/height (m)^2^.

Clinically confirmed pregnancy was defined as a vital after ultrasound confirmation of intrauterine pregnancy, with gestational age determined by ultrasound software in the first trimester of pregnancy. The blood was sampled, and quantitative analyses of serum TSH, fT4 and antithyroid antibodies, anti TPO Ab and anti Tg Ab, were performed with electro-chemiluminescent immunoassay, ECLIA (Roshe; Cobas blood 6000 e601 module) between 5 + 0 and 12 + 0 weeks of gestation (wg). TSH levels were measured at 8 + 6 weeks of gestation on average, the earliest at 5 + 1 weeks of gestation and the latest at 12 + 0 wg. TSH values were analyzed according to the trimester-specific reference range for gestation in the 2011 National Guideline of Good Clinical Practice for Thyroid Disorder, Republic of Serbia, and the European Thyroid Association (ETA) published in 2014 [15,33]. A TSH value of >2.5 µIU/mL in the first trimester of pregnancy with a normal fT4 value were the diagnostic criteria for subclinical hypothyroidism. According to the guidelines of the American Thyroid Association (ATA) from 2017, the diagnostic criterion for subclinical hypothyroidism is a TSH value of ≥4 µIU/mL [33]. ft4 values were analyzed according to a trimester-specific reference range. Based on available literature data (used in this study), the lower limit of the fT4 reference range in the first trimester on the 2.5th percentile as measured by immunoassay was around 10 pmol/L [16,25,34,35,36]. Antithyroid antibodies from pregnant women’s serum were analyzed by specific ECLIA immunoassay. The values of anti TPO Ab > 35 IU/mL and anti Tg Ab > 115 IU/mL were marked as the presence of autoimmune thyroid antibodies. If women were positive for at least one antibody type, they were considered to fulfill the diagnostic criteria for chronic thyroiditis [16,37].

The oral glucose tolerance test was performed between 24 + 0 and 30 + 0 wg with an average gestation of 26 to 27 weeks. The gestational diabetes mellitus diagnosis was established using the one-step strategy according to the WHO and IADPSG recommendations for the 2 h 75 g oGTT [5,6]. Diagnostic criteria were at least one value greater than fasting glucose of 5.1 mmol/L, a 1 h glucose of 10.0 mmol/L, or a 2 h glucose of 8.6 mmol/L. Following the oGTT test, the patients were divided into two groups: one with impaired glucose homeostasis, the GDM group, and the other with regular glucose metabolism, the group without GDM.

The association of elevated TSH values and the presence of antithyroid antibodies in the first trimester of pregnancy with impaired glycemic control was investigated. The relationship between different TSH values, both cut-offs > 2.5 µIU/mL and ≥4 µIU/mL, and GDM was examined. In addition, we examined the predictive value of elevated TSH levels and antithyroid antibody levels in the first trimester for detecting gestational diabetes mellitus. Model predictive performance for early prediction of GDM was evaluated by receiver operating characteristic curves.

### 2.3. Statistical Analysis

The results are presented as absolute and relative numbers (%), mean ± standard deviation or median (25th–75th percentile), depending on data type and distribution. Parametric (Student’s t test) and non-parametric (chi-square, Mann–Whitney U test, Fisher’s exact test) tests were used to assess the significance of the difference between the groups. Univariate and multivariate logistic regression were used to assess the significance of the GDM predictor. All tests with *p* values of <0.05 were considered statistically significant. Data were analyzed using SPSS 20.0 (IBM Corp.2011. IBM SPSS Statistics for Windows, version 20.0. Armonk, NY, USA: IBM Corp.).

## 3. Results

Out of 230 pregnant women, 19.5% (45) met the oGTT-75g criteria for GDM, and 80.5% (185) of the study population had normal glycemic control. There was no difference between the two groups in the examined characteristics, body height, body weight at the beginning and end of pregnancy, weight gain, and BMI, only in the mean age of patients (33.47 (±3.86) in the GDM group and 31.82 (±4.62) in non-GDM group, *p* = 0.028). The main characteristics of the examined group are shown in Table 1.

There was no difference in the family history of hereditary conditions, diabetes mellitus and thyroid disease in the first- and second-degree relatives between groups (Table 2).

There was no difference in the obstetric history of previous pregnancies. The observed difference in the prevalence of miscarriages and GDM in previous pregnancies between the two groups was not of statistical significance (Table 3).

The mean TSH value in the GDM group was significantly higher than in the group without GDM, 2.75 µIU/mL (±2.60) compared with 1.75 µIU/mL (±1.10), *p* = 0.027. The obtained fT4 values in both groups were in the reference range for the first trimester of pregnancy, although there was a significant difference (*p* = 0.021) between the examined groups: 14.05 (±2.38) in the group with GDM and 14.98 (±2.40) in the non-GDM group (Table 4).

Using the TSH > 2.5 IU/mL criterion, we found that 25.65% of the pregnant women were diagnosed with subclinical hypothyroidism, out of which one-third (8.26%) had impaired glucose tolerance. In the group with GDM, the prevalence of TSH > 2.5 µIU/mL was 42.2%. If we applied the ATA diagnostic criterion of TSH ≥ 4.0, SCH prevalence was 7.39%, with a prevalence of 3.48% among GDM. Among patients who were diagnosed with GDM, the prevalence of SCH was 17.78% (8 out of 45). The difference for both TSH values among the examined groups was statistically significant, 0.007 for TSH ≥ 4 IU/mL and 0.005 for TSH > 2.5 IU/mL (Table 5).

In this study, 25.6% of the pregnant women had at least one type of antithyroid antibody, which met the diagnostic criteria for chronic thyroiditis, and 10% had both types of antibodies. Hashimoto’s thyroiditis was diagnosed in 51.1% in the first trimester in the GDM group, compared to 19.46% in the non-GDM group, *p* < 0.001. The anti TPO Ab was detected in 19.6% of the pregnant women, and the anti Tg Ab in 15.2%. In the GDM group the anti TPO Ab was detected in 40%, while the anti Tg Ab was found in 26.67% (Table 6).

Elevated TSH ≥ 4 and both types of antibodies (anti TPO Ab and anti Tg Ab) were present in 4.44% of the GDM group and 1.62% in the non-GDM group. In the group of pregnant women with GDM, 13.3% had an elevated value of TSH ≥ 4 µIU/mL and at least one type of antibody detected in the first trimester, compared with 2.7% in the non-GDM group.

Modeling was performed using logistic regression analysis to estimate the predictive value of the examined parameters in the study. Several steps were used to assess the most important predictors of GDM. In the first step, all significant predictors obtained from the univariable analysis and variables deemed important by expert opinion (DM in family, BMI in the pregnancy onset, age in year) were used and evaluated for multivariable analysis. Using the enter method, all predictors with a *p* value of <0.200 were used for multivariable modeling. In this model, age, TSH and anti TPO Ab were significant, while anti Tg Ab was not (results not presented). Then, the backward method for variable reduction in the model with *p* < 0.10 probability was employed to eliminate non-significant predictors. The model revealed age, THS ≥ 4.0 µIU/mL and anti TPO Ab > 35 IU/mL as the most important and statistically significant predictors of GDM. Using the variance inflation factor, multicollinearity was evaluated. The area under the curve was used to evaluate the discriminative power of the model. Univariable and multivariable modeling are presented in Table 7.

Significant predictors of gestational diabetes mellitus were patient age, TSH ≥ 4 µIU/mL and anti TPO Ab > 35 IU/mL, as presented in Table 8.

## 4. Discussion

Thyroid disorders such as subclinical hypothyroidism have been associated with an increased risk of adverse pregnancy outcomes in most but not all published studies, probably because of the different criteria used to define elevated TSH levels [16,32,38,39]. TSH values in the first trimester of pregnancy normally decrease owing to the thyrotropic activity of increasing β-HCG, which has a homologous α subunit with TSH, and the free thyroxine (fT4) values increase at the same time.

In this study, the mean TSH values in the first trimester of pregnancy in GDM patients were higher than those in patients with normal glycemic control. Since 1953 in the Republic of Serbia, iodization of all salt intended for human consumption has been mandatory, so the assumption was that there were no pregnant women with iodine deficiency among the patients and that the thyroid dysfunction was not caused by that deficiency. There has been an increase in the number of patients with thyroid disorders, based on our clinical experience. To the best of our knowledge, this was the first study of its kind in the population of pregnant women in this area.

Elevated TSH levels may indicate a thyroid dysfunction that needs to be corrected to avoid maternal morbidity and to ensure normal fetal development, as well as numerous potential complications during pregnancy. Based on published studies, it has been estimated that 8–28% of pregnant women have a TSH concentration that is considered high if fixed TSH > 2.5 upper limits are used. Cut-offs set at these levels are too low and can result in overdiagnosis and overtreatment, which is unwarranted and may cause more harm than good [36,39]. New ATA guidelines from 2017 updated the recommendations by suggesting TSH values above 4.0 µIU/mL as a diagnostic criterion for subclinical hypothyroidism [16]. The percentage of patients in this study diagnosed with subclinical hypothyroidism using TSH values > 2.5 µIU/mL was 25.65%, which correlated with the findings presented in the T. Korevaar paper from 2018. That study demonstrated that with fixed TSH upper limits of 2.5 µIU/mL, 8–28% of pregnant women had a TSH concentration considered to be increased [36], in contrast to the significantly lower number of pregnant women (7.39%) diagnosed with SCH using the criterion TSH ≥ 4.0 µIU/mL. The number of patients diagnosed with GDM according to the guidelines of the WHO, IADPSG and the National Guideline for DM showed the expected growth trend, which was 19% in our study group [5,6,33].

Pregnant women with TSH > 2.5 µIU/mL in the first trimester were twice as likely to develop GDM, and pregnant women with TSH ≥ 4.0 µIU/mL were almost four times more likely to exhibit impaired glycemic control. This finding correlated with recent meta-analyses of cohort studies [40,41] and publications that suggested an association between elevated TSH and gestational diabetes mellitus [16,42]. Since thyroid hormones are involved in glucose metabolism, thyroid dysfunction during pregnancy in the form of subclinical hypothyroidism may be part of the chain of pathophysiological mechanisms in gestational diabetes mellitus onset.

The diagnosis of subclinical hypothyroidism in pregnancy is characterized by an fT4 value within the reference range for gestational age, although in this study the average value of fT4 was lower in the group of patients who developed GDM. That difference was statistically significant *p* = 0.02. This finding indicated the importance of the thyroid in glycemic control, so that lower fT4 values alone may affect the reduction in insulin sensitivity, increase insulin resistance, and result in hyperglycemia.

This research indicated a relatively high incidence of 25.6% of Hashimoto’s thyroiditis in the study population. Pregnant women who were diagnosed with at least one type of thyroid antibody above the threshold at the beginning of pregnancy, regardless of the presence of other thyroid function markers, had a three times higher risk of impaired glycemia. In this population of pregnant women anti TPO Ab was more commonly diagnosed than anti Tg Ab. If anti TPO antibodies were detected in the first trimester, there was a 2.7-fold higher risk of GDM, and if anti Tg Ab was detected, there was twice the risk of GDM compared to pregnant women who had no antibodies present.

Pregnant women diagnosed with GDM were almost five times (4.9) more likely to have elevated TSH ≥ 4.0 µIU/mL and at least one type of antibody present in the titer as diagnostic criteria for autoimmune thyroiditis, compared to pregnant women with normal glycemic control.

Increased TSH levels and the presence of thyroid antibodies showed a positive correlation with the risk of GDM. The corresponding odds values for TSH and anti TPO Ab were OR 3.971, 95% CI: 1.34–11.77 and OR 4.026, CI: 1.866–8.689, respectively.

Through multivariate logistic regression, we demonstrated that significant predictors of gestational diabetes mellitus that may improve first-trimester pregnancy screening performances are patient age, TSH ≥ 4 µIU/mL and anti TPO Ab > 35 IU/mL. The receiver operating characteristic (ROC) curve for significant variables achieved an area under the curve (AUC) of 0.711; 95% confidence interval (CI) 0.629–0.793.

Increased risk of GDM in pregnant women with positive anti TPO Ab in early pregnancy and a higher rate of positive anti TPO Ab in pregnant women with GDM than in those without GDM have already been shown in publications, and that finding correlated with this study [35,43].

Insulin resistance, defined as the inability of insulin to increase glucose uptake and utilization in peripheral tissues, can be present for years before the onset of hyperglycemia, and it is known that hypothyroidism is associated with decreased insulin sensitivity since thyroid hormones directly regulate insulin secretion. The link between impaired glycemia and thyroid dysfunction may be related to an autoimmune mechanism that occurs as part of the polyglandular autoimmune syndrome, involving both the pancreas and the thyroid gland.

This study did not examine the presence of anti-insulin antibodies to prove this association, as they are an autoimmune marker of pancreatic islet cell beta cell destruction [44] and can be detected years before clinical manifestations of DM in healthy individuals or in individuals with other autoimmune disorders such as autoimmune thyroiditis. Some authors believe that the link between the two is an autoimmune inflammatory process, citing elevated serum cytokine levels in both disorders [35,45,46]. Hashimoto’s disease could be considered a T helper 1 disease in which pro-inflammatory cytokines play a crucial role, and there is a connection between the level of anti TPO antibodies and pro-inflammatory cytokines that appears at higher concentrations in individuals with insulin resistance [47].

Further research is needed to better define the relationship between these two disorders in pregnant women and to better understand their pathogenetic mechanisms, allowing us to find the best possible predictor of forthcoming disorders.

Since there are many possible short and long-term complications of gestational diabetes mellitus in the mother and fetus, there is a clear need for early detection of glycemic disorders and, accordingly, adequate treatment to prevent and reduce the consequences of the disorder.

The current risk factor-based screening method estimates GDM risk by considering maternal characteristics and personal and family history. Clinical screening markers have limited diagnostic accuracy when used separately. Prediction models that include a combination of different markers may improve the sensitivity and specificity of GDM screening by risk factors. The thyroid biochemical screening markers can be used as additional markers in the first trimester since they are easily accessible and feasible. The clinical markers currently employed may be combined with thyroid markers, TSH and anti TPO Ab to improve screening performance. Numerous risk estimation models for GDM can be found in the current literature; however, most of them are based on different diagnostic criteria. Validation of predictive models in clinical practice is crucial in large cohorts and across different ethnic groups [48]. Further research is needed to confirm the benefit and effectiveness of screening strategies in the detection of GDM and prevention and reduction in perinatal complications.

There are limitations in this study that must be mentioned. The study was monocentric with a relatively small number of observation units. All patients included in our study were from the Republic of Serbia, with a certain specific genetic heritage that may differ in other populations. The research was conducted in the population of pregnant women under regular antenatal care in a tertiary institution to which patients with high-risk pregnancy gravitate, so it is possible that the actual prevalence of thyroid disease and gestational diabetes is slightly lower than in our study.

## 5. Conclusions

Numerous studies have evaluated new markers for first-trimester pregnancy screening to predict pregnancy complications and prevent adverse outcomes. This study suggested an association between elevated TSH values and the onset of hyperglycemia. Autoimmune thyroiditis was shown as an independent marker for GDM occurrence. According to the results of this research, elevated TSH values and anti TPO antibodies in early pregnancy might be used as additional first-trimester markers to improve screening performance for gestational diabetes mellitus.

## Figures and Tables

**Table 1 jcm-11-03916-t001:** Maternal parameter characteristics of the study population.

Parameters	GDM Group (n = 45)	Non-GDM Group (n = 185)	*p* Value
age (years) *(MV ± SD)*	33.47 (±3.86)	31.82 (±4.62)	**0.028**
body height *(MV ± SD)* (cm)	168.22 (±4.61)	168.74 (±6.75)	0.540
b.w. at the pregnancy start *(MV ± SD)* (kg)	66.67 (±9.82)	64.90 (±13.19)	0.401
b.w. at delivery *(MV ± SD)* (kg)	78.98 (±11.27)	77.41 (±11.48)	0.422
weight gain *(MV ± SD)* (kg)	12.80 (±6.39)	12.92 (±4.79)	0.904
BMI *(MV ± SD)* (kg/m^2^)	23.72 (±3.65)	22.79 (±4.14)	0.171

Data presented as *MV* ± *SD*; *MV*: mean value; *SD*: standard deviation; GDM: gestational diabetes mellitus; b.w.: body weight; BMI: body mass index.

**Table 2 jcm-11-03916-t002:** Family history of hereditary conditions.

		Non-GDM (n = 185)	GDM (n = 45)	*p* Value
Family history of thyroid disorders *N* (*N*%)	no	150 (79.8%)	38 (20.2%)	0.647
	yes	35 (82.9%)	7 (17.1%)	
Family history of diabetes mellitus *N* (*N*%)	no	112 (79.4%)	29 (20.6%)	0.511
	yes	73 (83.0%)	15 (17.0%)	

Data presented as number *N* and percentage (%); GDM: gestational diabetes mellitus.

**Table 3 jcm-11-03916-t003:** Previous obstetric history.

		Non-GDM (n = 185)	GDM (n = 45)	*p* Value
Deliveries *N* (*N*%)	no	98 (79.7%)	5 (20.3%)	0.782
	yes	86 (81.1%)	20 (18.9%)	
Miscarriages *N* (*N*%)	no	135 (81.8%)	30 (18.2%)	0.369
	yes	49 (76.6%)	15 (23.4%)	
GDM in previous pregnancy	no	77 (89.54%)	16 (75.0%)	0.241
	yes	9 (10.46%)	4 (25.0%)	

Data presented as number *N* and percentage (%); GDM: gestational diabetes mellitus.

**Table 4 jcm-11-03916-t004:** Biochemical parameters in the study population.

Parameters	GDM(n = 45)	Non-GDM (n = 185)	*p* Value
Fasting glucose (mmol/L) *(MV ± SD)*	4.79 (±0.53)	4,37 (±0.37)	
Insuin fasting * (uIU/mL) *M (IQ)*	8.00 (5.3–11.5)	8.81 (6.6–11.8)	0.191
TSH (µIU/mL) *(MV ± SD)*	2.75 (±2.60)	1.75 (±1.10)	**0.027**
fT4 (pmol/L) *(MV ± SD)*	14.05 (±2.38)	14.98 (±2.40)	**0.021**

Data presented as *MV ± SD*; MV: mean value; SD: standard deviation; * Insulin levels expressed as M (IQ); M: median; IQ: interquartile range; GDM: gestational diabetes mellitus; TSH: thyroid-stimulating hormone; fT4: free thyroxine.

**Table 5 jcm-11-03916-t005:** The prevalence of elevated TSH in the study population.

Parameters	GDM Group (n = 45)	Non-GDM Group (n = 185)	*p* Value
TSH ≥ 4.0 µIU/mL *N* (*N*%)	8 (17.8%)	9 (4.86%)	**0.007**
TSH > 2.5 µIU/mL *N* (*N*%)	19 (42.2%)	40 (21.6%)	**0.005**

Data presented as number *N* and percentage (%); GDM: gestational diabetes mellitus; TSH: thyroid-stimulating hormone.

**Table 6 jcm-11-03916-t006:** The prevalence of antithyroid antibodies in the study population.

Parameters	GDM Group (n = 45)	Non-GDM Group (n = 185)	*p* Value
Hashimoto thyroiditis *N* (*N*%)	23 (51.10%)	36 (19.46%)	**<0.001**
Anti TPO Ab > 35 IU/mL *N* (*N*%)	18 (40.00%)	27 (14,59%)	**<0.001**
Anti Tg Ab > 115 IU/mL *N* (*N*%)	12 (6.67%)	23 (12.43%)	**0.017**

Data presented as number *N* and percentage (%); GDM: gestational diabetes mellitus; anti TPO Ab: anti-thyroid peroxidase antibodies; anti Tg Ab: anti-thyroglobulin antibodies.

**Table 7 jcm-11-03916-t007:** Univariable and multivariable modeling.

	Univariable		Multivariable (Backward) *	
	OR (95% CI)	*p* Value	OR (95% CI)	*p* Value
**Age (yrs)**	1.084 (1.008–1.167)	0.030	**1.100** (1.017–1.189)	**0.017**
Age 35+	1.399 (0.709–2.759)	0.333		
BMI (kg/m^2^)	1.053 (0.977–1.135)	0.174		
DM in family	0.794 (0.398–1.581)	0.511		
**TSH** **≥ 4 µIU/mL**	4.228 (1.530–11.581)	0.005	**2.962** (0.992–8.846)	**0.052**
TSH ≥ 2.5 µIU/mL	2.649 (1.332–5.267)	0.005		
Hashimoto Thyroiditis	4.327 (2.173–8.614)	<0.001		
**Anti TPO Ab > 35** **IU/mL**	3.901 (1.894–8.036)	<0.001	**3.627** (1.682–7.821)	**0.001**
Anti Tg Ab > 115 IU/mL	2.561 (1.160–5.655)	0.020		

* AUC of the model = 0.711 (95% CI, 0.629–0.793). BMI: body mass index; DM: Diabetes mellitus; TSH: thyroid-stimulating hormone; anti TPO Ab: anti-thyroid peroxidase antibodies; GDM: gestational diabetes mellitus; anti Tg Ab: anti-thyroid globulin antibodies.

**Table 8 jcm-11-03916-t008:** Significant predictors of gestational diabetes mellitus.

	OR	*p* Value
AGE (years)	1.100	**0.017**
TSH ≥ 4 µIU/mL	2.962	**0.052**
Anti TPO Ab > 35 IU/mL	3.627	**0.001**

TSH: thyroid-stimulating hormone; anti TPO Ab: anti-thyroid peroxidase antibodies; GDM: gestational diabetes mellitus.

## Data Availability

Original data are available on request.

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
