# Peer review of "Can Thyroid Screening in the First Trimester Improve the Prediction of Gestational Diabetes Mellitus?"

_jcm, 2022, doi:10.3390/jcm11133916_

Round 1

Reviewer 1 Report

Dear Authors,

my comments:

  1. In my opinion an English editing is essential.
  2. You have to focus on one type of English- in text I found two styles- American English and British English
  3. All abbreviations should be described and checked one more time. e.g. in one place you wrote "GDM" in other "DMG" describing the same thing.
  4. Legends in tables 1, 2, 3, 4,6 need corrections.
  5. titles of the tables should be over the tables not under.
  6. The all thext should be performed in one style of technical writing.. I see 3 styles...
  7. In my opinion in conclusion the word "can" in "...can be use as additional first-trimester markers" is too big word.. maybe "might" is better.
  8. Not enough number of recent articles in references 

Author Response

To Reviewer 1

We are grateful for taking your time to review our scientific paper.

Thank you very much for reviewing the manuscript and providing all the suggestions for improvements. Hopefully, our paper will be considered for publication in the journal after the changes we made in response to your suggestions.

  1. Manuscript submitted to MDPI English language editing service.
  2. Manuscript submitted to MDPI English language editing service
  3. All abbreviations have been checked and corrected.
  4. Legends in all tables have been checked and corrected.
  5. The position of the table titles was adjusted as suggested.
  6. Manuscript submitted to MDPI English language editing service
  7. Corrected as suggested: According to the results of this research, elevated TSH values and anti TPO antibodies in early pregnancy might be used as additional first-trimester markers to improve screening performance for gestational diabetes mellitus. (marked in red)
  8. The most recent articles as the reference have been added in the introduction and in the discussion.

Reviewer 2 Report

Dear Authors,

This paper investigates an important topic, however the correlation between maternal thyroid disorders and gestational diabetes has already been described in the literature, as the current manuscript states. I would like to suggest the following comments. Thank you for considering my feedback.  

Abstract and Introduction: can the authors please clearly state the research gap they aim to address with the study? This will clarify what this study adds to the previous literature.

Introduction: Given that the authors are focusing on a predicting model to identify pregnant women at highest risk for a maternal complication (GDM) rather than a fetal condition, the description of first-trimester screening for chromosomal abnormalities may not be very relevant here. Instead, they may discuss about first trimester screening for another maternal condition (such as pre-eclampsia).

Materials and Methods: The authors state that all pregnant women had first trimester screening to detect fetal congenital abnormalities and that fetal malformations were an exclusion criteria. Given that the majority of fetal abnormalities may be detected only after the first trimester, did the authors collect information of the anatomy ultrasound of the second trimester? Please clarify.  

Study population size: please clarify whether the initial population size of 260 patients was based on power calculations or not.

Table 2: Please consider adding the frequency of previous pregnancy complicated by GDM and/or thyroid disorders. Or are these included in “Family history”? Please clarify, and, if possible, report previous obstetric history as separated from family history.

Table 6: This is the most important table for the presented study. However, it is not very clear how the predictive model was built. Please clarify the covariates included in the analysis. Also, given the overall large range of gestational age at testing, for both thyroid labs (5-12 weeks) and oral glucose test (24-20 weeks), did the authors control for gestational age at testing (both thyroid and oral glucose testing)?

Author Response

To Reviewer 2

Thank you so much for your time reviewing this article as well as all the suggestions to improve the manuscript. Since I am in perinatology field, I am very happy that you recognize the importance of screening in perinatology. I hope that, after the changes were made, according to your remarks, our paper might now be considered for publication in the journal.

  1. I hope that I clarified the aim and the difference to the previous publications by adding the paragraph: Existing screening tests for gestational diabetes mellitus use maternal characteristics, family, and personal history to assess the risk of one specific condition [1]. The possibility of using screening markers of thyroid function in the early first trimester of pregnancy to assess the risk of gestational diabetes mellitus might allow one test to identify the risk for two clinically important conditions, enabling appropriate and specific treatment for pregnant women at risk (marked in red).
  2. Writing about the first trimester screening for chromosomal abnormalities, the idea was to make a point that the implementation of the first trimester screening was a revolutionary milestone in the pregnancy monitoring. Your remark is valuable, since the article is focusing on maternal condition complication. According to your suggestion, the paragraph about the preeclampsia screening in the first trimester added in the introduction, to clarify the aim of screening for the maternal condition complication during the pregnancy: Recent studies have shown that certain maternal conditions may be predicted in the first trimester. Screening a combination of maternal factors and measuring mean arterial pressure, uterine pulsatility index, and serum placental growth factor may predict 90% of early preeclampsia cases, before 32 weeks, and 75% of preterm preeclampsia cases, before 37 weeks, with a 10% false-positive rate [1] (marked in red).
  3. During the research and collecting the data, all the pregnant women were regularly followed. To better explain the procedures paragraph changed and added: All the pregnant women were followed-up every 4 to 6 weeks, had at least five to six visits and underwent thyroid function screening in the first trimester and an oral glucose tolerance test (oGTT) in the late second or early third trimester of pregnancy. The first visit occurred between 5 and 11 weeks of pregnancy to determine the exact date of the pregnancy, perform the viability ultrasound scan, obtain data on the mother and her family history, perform weight and height measurements, and administer a thyroid function screening test. At the second visit, between 11 and 14 gestational weeks, first-trimester screening for chromosomal abnormality was carried out to assess the risk of chromosomal abnormality, according to the standards of the Fetal Medicine Foundation as part of the international and national guidelines, and detect major congenital anomalies as early as possible. A detailed anatomy ultrasound scan between 20 and 24 gestational weeks was carried out on every pregnant woman as the standard of pregnancy monitoring, to check growth and exclude structural anomalies. The oGTT with 75 g, was performed between 24 and 30 gestational weeks. Growth and well-being ultrasound scans were performed between 28 and 32 gestational weeks and were repeated between 36 and 38 gestational weeks, if the baby was not delivered earlier.
  4. We tried to collect data from as many responders for the research, although we have had the statistical calculation of the minimum cohort before the research has started. To clarify the explanation added in the Study setting and Patients: The minimum number of respondents for estimating the prevalence of gestational di-abetes in pregnant women, calculated based on literature data indicating a 1-14% prev-alence of gestational diabetes (an assumption that in our population (in Serbia) is about 10%), with an alpha error of 0.05 and an accuracy of 5%, was 138. Considering the pos-sibility of 10% missing data, the required minimum number of respondents for this research was 152. The study enrolled 260 patients. Twenty-five patients were excluded because of incomplete documentation, one withdrew owing to early fetal demise (miscarriage), and four patients left the study. The trial covered 230 patients.
  5. Two separate tables made one for family history of hereditary conditions and second previous obstetric history, GDM in previous pregnancy has added. The patients with previous thyroid disorders excluded by studies inclusion criteria

  1. Multivariable logistic regression model with significant predictors of GDM, presented in different manner.

OGTT with 75g, was usually done between 24 and 30 gestational weeks to avoid to miss the diagnosis of GDM, with an average gestation of 26 to 27 weeks. TSH levels were measured at 8+6 weeks of gestation on average, as early as 5+1 weeks of gestation and the latest at 12+0 gestational weeks.There is the record for every pregnant woman with exact gestational age at testing (thyroid screening and OGTT)

Reviewer 3 Report

Milovanovic and colleagues  address an interesting topic. Results suggest that the presence of altered thyroid function laboratory results may be accompanied by other metabolic alterations such as insulin resistance or gestational diabetes. The document emphasizes the importance of prenatal care. I have some comments for authors to address before considering its publication. 

1.     Proofread the entire manuscript, it will benefit from English editing. 

2.     Page 3/Procedures: It is important to make a distinction between self-reported pre-pregnancy weight and BMI vs. weight and BMI measured at first prenatal visit. 3.     What was the mean follow-up time for these patients? How many visits did they complete in the study? Could you specify when the laboratory tests were performed?

4.     Table 2: Some numbers do not add correctly in the table e.g. number of total miscarriages in Non GDM, number of deliveries. 

5.     Page 7, Lines 230-231 “which correlates with the data presented in the paper from 2018.” Could you please specify the paper?

6.     Page 7, Lines 232-235: Please add a reference “The number of patients diagnosed with GDM according to the guidelines of the WHO, IADPSG 233 and the National Guideline for DM shows the expected growth trend,”

Author Response

To reviewer 3

Thank you so much for your time reviewing this article as well as all the suggestions to improve the manuscript. Since I am in the field of perinatology, I am very happy that you recognize the importance of screening for maternal condition complications during pregnancy and understand the complicated relationship between different conditions in perinatology. I hope that, after the changes were made, according to your remarks, our paper might now be considered for publication in the journal. 

  1. Manuscript submitted for language editing
  2. There was the mistake in writing. Body height and weight of pregnant women were measured at the first visit (and, on each visit) and used to calculate the body mass index (BMI) at the pregnancy start, NOT pre pregnancy. Corrected in the paragraph Procedures, and in the table No 1.
  3. The mean follows up time was 4 to 6 weeks, The pregnant women have at least 5 to 6 visits. To better explain the procedures paragraph changed and added: All the pregnant women were followed-up every 4 to 6 weeks, had at least five to six visits and underwent thyroid function screening in the first trimester and an oral glucose tolerance test (oGTT) in the late second or early third trimester of pregnancy. The first visit occurred between 5 and 11 weeks of pregnancy to determine the exact date of the pregnancy, perform the viability ultrasound scan, obtain data on the mother and her family history, perform weight and height measurements, and administer a thyroid function screening test. At the second visit, between 11 and 14 gestational weeks, first-trimester screening for chromosomal abnormality was carried out to assess the risk of chromosomal abnormality, according to the standards of the Fetal Medicine Foundation as part of the international and national guidelines, and detect major congenital anomalies as early as possible. A detailed anatomy ultrasound scan between 20 and 24 gestational weeks was carried out on every pregnant woman as the standard of pregnancy monitoring, to check growth and exclude structural anomalies. The oGTT with 75 g, was performed between 24 and 30 gestational weeks. Growth and well-being ultrasound scans were performed between 28 and 32 gestational weeks and were repeated between 36 and 38 gestational weeks, if the baby was not delivered earlier.
  4. TSH levels were measured at 8+6 weeks of gestation on average, as earliest at 5+1 weeks of gestation and the latest at 12+0 gestational week.

OGTT with 75g, was done between 24 and 30 gestational weeks, with an average gestation of 26 to 27 weeks

There is the record for every pregnant woman with exact gestational age at testing (thyroid screening and OGTT)

  1. The number in the table No 2 corrected.
  2. The publication from 2018 is T. Korevaar. The upper limit for TSH during pregnancy: why we should stop using fixed limits of 2.5 or 3.0 mU/l. Explained in the text as sugested: The percentage of patients in this study diagnosed with subclinical hypothyroidism using TSH values >2.5 µIU/ml was 25.65%, which correlated with the findings presented in the T. Korevaar paper from 2018. That study demonstrated that with fixed TSH upper limits of 2.5 µIU/ml, 8–28% of pregnant women had a TSH concentration considered to be increased [36], in contrast to the significantly lower number of pregnant women (7.39%) diagnosed with SCH using the criterion TSH ≥4.0 µIU/ml. The refference added in the text.

Round 2

Reviewer 1 Report

Dear Authors,

I accept your reply.